# SkillBlender: Towards Versatile Humanoid Whole-Body Loco-Manipulation via Skill Blending

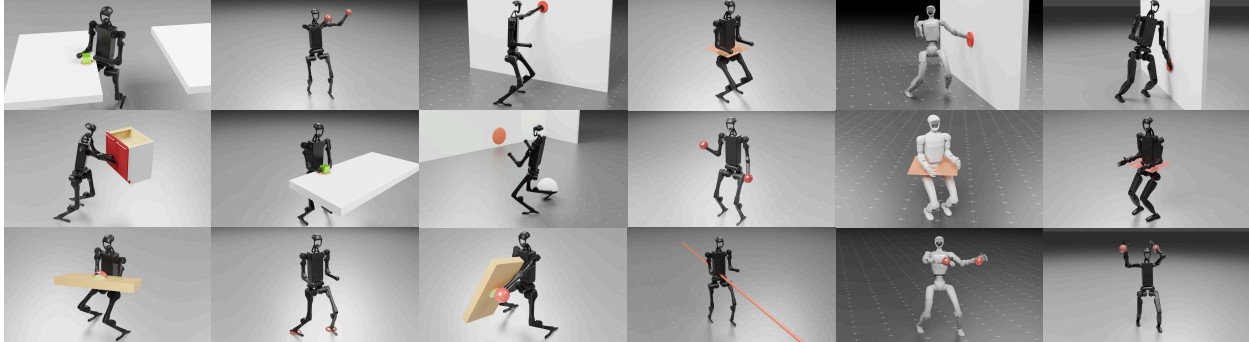

Figure 1: **SkillBlender** performs **versatile** autonomous humanoid loco-manipulation tasks within different embodiments and environments, given **only one or two** intuitive reward terms.

## Abstract

Humanoid robots hold significant potential in accomplishing daily tasks across diverse environments thanks to their flexibility and human-like morphology. Recent works have made significant progress in humanoid whole-body control and loco-manipulation leveraging optimal control or reinforcement learning. However, these methods require tedious task-specific tuning for each task to achieve satisfactory behaviors, limiting their versatility and scalability to diverse tasks in daily scenarios. To that end, we introduce **SkillBlender**, a novel hierarchical reinforcement learning framework for **versatile** humanoid loco-manipulation. SkillBlender first pretrains goal-conditioned task-agnostic primitive skills, and then dynamically blends these skills to accomplish complex loco-manipulation tasks **with minimal task-specific reward engineering**. We also introduce **SkillBench**, a parallel, cross-embodiment, and diverse simulated benchmark containing three embodiments, four primitive skills, and eight challenging loco-manipulation tasks, accompanied by a set of scientific evaluation metrics balancing accuracy and feasibility. Extensive simulated experiments show that our method significantly outperforms all baselines, while naturally regularizing behaviors to avoid reward hacking, resulting in more accurate and feasible movements for diverse loco-manipulation tasks in our daily scenarios. Our code and benchmark will be open-sourced to the community to facilitate future research. Anonymous project page: https://skillblender-anon.github.io/.

## 1 Introduction

Humanoid robots hold significant potential to be seamlessly deployed in our daily lives to accomplish everyday tasks across diverse environments due to their flexibility and human-like morphology. This alignment is crucial since our environments, tasks, and tools are designed around human capabilities (Sferrazza et al., 2024). Specifically, we want humanoids to perform versatile loco-manipulation tasks in our daily lives autonomously, instead of executing pre-programmed motions only. However, humanoid loco-manipulation remains extremely

challenging due to the high-dimensional nature of their observation and action spaces, as well as the complex dynamics inherent in bipedal systems (Hurmuzlu et al., 2004). Previous optimal control-based works focused on building dynamic models for model predictive control (MPC) (Gazar et al., 2021; Khazoom et al., 2024), which have made great progress on humanoid control. On the other hand, recent model-free reinforcement learning (RL) methods (Gu et al., 2024; Cheng et al., 2024; Zhang et al., 2024a; Fu et al., 2024; He et al., 2024b;a) have also made significant strides in agile humanoid whole-body control, benefiting from highly parallel simulation training (Makoviychuk et al., 2021; Schulman et al., 2017) that largely improves sample efficiency.

However, these methods fall short of making humanoid robots **versatile** — being able to perform diverse tasks in a scalable way. For instance, optimal control-based methods often require building and tuning accurate dynamic models and complex cost functions tailored to every single task, along with time-intensive optimization, limiting their scalability across different tasks. Regarding RL-based methods, first, most of them are mainly designed for expressive tasks like motion mimicking or teleoperation, lacking the ability to perform versatile loco-manipulation tasks autonomously. Moreover, to successfully learn a task and avoid reward hacking, RL-based methods often require labor-intensive reward shaping to balance terms like task success, orientation, upper body pose, gait, contact, curiosity, etc. (Gu et al., 2024; Cheng et al., 2024; Zhang et al., 2024a; van Marum et al., 2024) for each single task, limiting their versatility and possibility for infinite task variations in our daily lives. Therefore, it's crucial to find a painless and scalable way for humanoids to learn versatile loco-manipulation capabilities without extensive task-specific tuning.

To that end, we draw inspiration from human motor skill development, where fundamental capabilities like walking and reaching are acquired first and later combined for more complex tasks (Ruffin, 2009), enabling sophisticated whole-body coordination. By leveraging these skill priors, we humans can perform versatile tasks in our daily lives without learning them from scratch. To mimic such mechanisms, we propose **SkillBlender**, a novel Hierarchical Reinforcement Learning (HRL) framework for **versatile** humanoid whole-body loco-manipulation, leveraging a pretrain-then-blend paradigm **with minimal reward engineering**. We first pretrain a set of goal-conditioned primitive skills that are task-agnostic, reusable, and physically interpretable. A high-level controller then learns to synthesize subgoals and per-joint weight vectors to blend these low-level skills. Unlike prior HRL methods (Peng et al., 2019; Yang et al., 2020; Kumar et al., 2023), our approach proposed a unique vectorized weighting mechanism to blend different skills, enabling more flexible and accurate humanoid actions. This hierarchical structure not only simplifies the search space when training the high-level controller but also reduces the need for extensive reward engineering, requiring **only one or two** reward terms per task. This enables our method's versatility, generality, and scalability to diverse loco-manipulation tasks in our daily scenarios.

Beyond our algorithmic contributions, we recognize the critical role of simulation benchmarks in humanoid learning (Geng et al., 2025). Many previous benchmarks either do not support fully parallel simulation (Sferrazza et al., 2024; Al-Hafez et al., 2023; Tassa et al., 2018) or lack whole-body coordination (Chernyadev et al., 2024). More importantly, they overlook the humanoid's motion feasibility, which encourages reward hacking (Hansen et al., 2024) to maximize task returns, leading to unnatural or unrealistic behaviors if without careful reward engineering (Sferrazza et al., 2024; Liu et al., 2024). This underscores the need for a **parallel, comprehensive, and scientifically grounded** benchmark to systematically evaluate humanoid loco-manipulation, balancing task accuracy and motion feasibility. As such, we also introduce **SkillBench**, a parallel, cross-embodiment, and diverse benchmark for humanoid whole-body loco-manipulation. SkillBench supports three distinct humanoid morphologies, four primitive skills, and eight challenging loco-manipulation tasks. Unlike previous benchmarks that only assess success via task return (Tassa et al., 2018; Al-Hafez et al., 2023; Sferrazza et al., 2024), our evaluation framework incorporates metrics from two complementary dimensions: (1) the accuracy metric to measure task completion success and (2) a set of feasibility metrics to assess the naturalness and realism of humanoid motion.

Our extensive experiments on our simulated benchmark show that our SkillBlender significantly outperforms existing baselines in both accuracy and feasibility, producing more natural and feasible behaviors with minimal task-related rewards. Our code and benchmark will be open-sourced to promote future research. In summary, the key contributions of our work are three-fold:

- We propose **SkillBlender**, a pretrain-then-blend framework for versatile humanoid whole-body loco-manipulation. With our unique skill blending strategy, SkillBlender produces more accurate and feasible behaviors for diverse loco-manipulation tasks with minimal reward engineering.
- We introduce **SkillBench**, a parallel, cross-embodiment, and diverse benchmark for humanoid whole-body loco-manipulation for comprehensive evaluation. Our benchmark includes two complementary metrics that measure both the accuracy and feasibility of humanoid motions.
- We provide and will open-source a set of structured, broadly useful, reusable, and task-agnostic humanoid primitive skills and diverse simulated task environments, as well as models and pretrained checkpoints to facilitate future open humanoid research.

## 2 Related Works

**Humanoid Whole-Body Control** Humanoid whole-body control and loco-manipulation remains extremely difficult due to its high dimensionality and unstable bipedal nature. To tackle this problem, previous non-learning-based methods focused on building dynamic models for MPC (Gazar et al., 2021; Khazoom et al., 2024). However, these methods require tuning accurate dynamic models and complex cost functions for each task, along with time-consuming optimization. Recent times witnessed significant progress on learning-based methods leveraging model-free reinforcement learning for humanoid locomotion (Radosavovic et al., 2024; Gu et al., 2024; Zhuang et al., 2024), motion tracking (Cheng et al., 2024; Fu et al., 2024; He et al., 2024b;a;c; Ji et al., 2024; He et al., 2025a; Ben et al., 2025; Li et al., 2025; Ze et al., 2025), loco-manipulation (Zhang et al., 2024a), and other tasks (Huang et al., 2025; He et al., 2025b; Zhuang & Zhao, 2025), due to RL's robustness against model mismatch and uncertainties, as well as capabilities of real-time agile motions on legged robots (Lee et al., 2020; Miki et al., 2022). However, most of them only focused on locomotion or motion mimicking, lacking the abilities to autonomously perform versatile loco-manipulation tasks in a scalable manner. Moreover, these RL-based methods require lots of tedious reward tuning on orientation, gait, contact, curiosity, etc., on each setting (van Marum et al., 2024), limiting their versatility and possibility for infinite task variations in our daily lives. Compared to these works, our **SkillBlender** overcomes the need for tedious reward engineering, generally requiring up to two reward terms for each task to train robust and natural policies for versatile humanoid loco-manipulation without any motion priors, which is scalable to diverse autonomous loco-manipulation tasks.

**Hierarchical Reinforcement Learning** Hierarchical Reinforcement Learning (HRL) strategies have been used in many works to handle the complex temporal dependencies of long-horizon tasks, which are challenging for conventional RL (Sutton et al., 1999; Bacon et al., 2017; Heess et al., 2016; Li et al., 2020; Escontrela et al., 2022). HRL has also seen frequent application in quadruped loco-manipulation (Yang et al., 2020; Kumar et al., 2023; Ji et al., 2022; Cheng et al., 2023; Zhang et al., 2024b) and physics-based animation (Starke et al., 2019; Peng et al., 2019; Won et al., 2020; Wang et al., 2020; Peng et al., 2022; Luo et al., 2023a;b; Wang et al., 2024). Recently Sferrazza et al. (2024); Hansen et al. (2024) have also shown promising results of HRL on humanoid whole-body control to boost policy learning and avoid reward hacking. However, those methods only consider a single low-level policy instead of multiple reusable skills which are more structural for complex whole-body loco-manipulation tasks. Compared to MCP (Peng et al., 2019) or ASE (Peng et al., 2022) which consider multiple skills, our method's low-level skills are goal-conditioned and physically interpretable, which are specialized and generally useful, allowing them to be reused and blended effectively. Additionally, our **SkillBlender** proposed a unique skill blending strategy with vectorized weighting, allowing more flexible, accurate, and feasible movements.

**Humanoid Learning Benchmarks** Due to the sheer complexity of humanoid robots, it is essential to build benchmarks for humanoid whole-body loco-manipulation, especially simulated benchmarks due to the expense and danger of humanoid hardware. Many previous benchmarks either focus exclusively on locomotion (Tassa et al., 2018), consider loco-manipulation to a limited extent (Al-Hafez et al., 2023), or are geared towards animation with virtual animation characters (Tassa et al., 2018). They do not support parallel simulation either, which is extremely important for RL training. Recently, HumanoidBench (Sferrazza et al., 2024) began benchmarking loco-manipulation on actual robot models; however, its parallelization capabilities remain limited, supporting only a small number of parallel environments if not disabling most

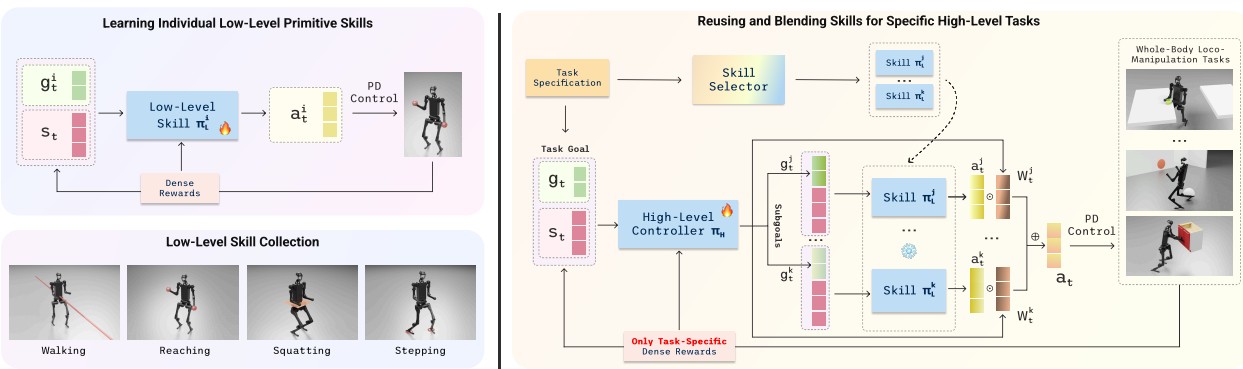

Figure 2: Overview of **SkillBlender**. We first pretrain goal-conditioned primitive expert skills that are task-agnostic, reusable, and physically interpretable, and then reuse and blend these skills to achieve complex whole-body loco-manipulation tasks given **only one or two** task-specific reward terms.

collisions. BiGym (Chernyadev et al., 2024) leverages a Unitree H1 robot to benchmark a variety of bimanual manipulation tasks; however, it uses a floating base for all the demonstrations that fall out of whole-body loco-manipulation. Recent Mimicking-Bench (Liu et al., 2024) is mainly used for human motion tracking purposes with limited embodiments and tasks. Moreover, all these previous benchmarks lack scientific metrics to evaluate motion feasibility and naturalness. In this work, we attempt to address the aforementioned limitations by introducing our massively parallel, cross-embodiment, and diverse **SkillBench** to systematically benchmark humanoid whole-body loco-manipulation algorithms with scientific evaluations.

## 3 SkillBlender

Our **SkillBlender** draws inspiration from human growth and development: infants lift and turn their heads before they can turn over, and move their limbs (arms and legs) before grasping an object (Ruffin, 2009). As humans, we first learn individual primitive motor skills when we grow up. These primitive skills are generalized, reflexive, and task-agnostic so that they are not tied to specific tasks and can be used in various daily tasks, and they also have physical goals like walking to one specific location, reaching a specific target, etc. When we encounter a new task, like dancing, that requires whole-body coordination, we can then reuse and blend these skills to finish the task.

Inspired by this, we propose our method SkillBlender, which first pretrains a set of goal-conditioned primitive skills, that are task-agnostic, reusable, and physically interpretable, and then dynamically blends these skills given a new high-level task, requiring minimal reward engineering. In this section, we will describe our problem formulation in Sec. 3.1, low-level primitive skill learning in Sec. 3.2, and high-level skill blending in Sec. 3.3.

### 3.1 Problem Formulation

We formulate our humanoid whole-body loco-manipulation policy learning problem as a goal-conditioned Markov Decision Process (MDP) $\mathcal{M} = \langle \mathcal{S}, \mathcal{A}, \mathcal{T}, \mathcal{R}, \gamma, \mathcal{G} \rangle$ of state $s \in \mathcal{S}$, action $a \in \mathcal{A}$, transition function $\mathcal{T}$, reward $r \in \mathcal{R}$, discount factor $\gamma$, and task goal $g \in \mathcal{G}$. The objective is to maximize the expected return $\mathbb{E}\left[\sum_t \gamma^t r_t\right]$ by finding an optimal policy $\pi^*(a_t|g_t, s_t)$, where the subscript $t$ indexes the time step.

In our hierarchical pipeline, we label the $i$-th low-level primitive skill as $\pi_L^i(a_t^i|g_t^i, s_t)$, which is responsible for outputting humanoid actions $a_t^i$ conditioned on the skill's subgoal $g_t^i$ and the current state $s_t$. We label the high-level controller as $\pi_H(\{g_t^i\}, \{W_t^i\}|g_t, s_t)$, which outputs a subgoal $g_t^i$ and weight vector $W_t^i$ for each low-level skill conditioned on the high-level goal $g_t$, and state $s_t$.

### 3.2 Learning Individual Low-Level Primitive skills

At the core of our framework is the ability to reuse and blend goal-conditioned primitive skills for new tasks, requiring only minimal task-specific reward tuning with only one or two reward terms. To achieve this, we first pretrain a set of low-level primitive skills as whole-body policies, denoted as $\pi_L^i$, using goal-conditioned reinforcement learning. More specifically, each low-level skill policy $\pi_L^i(a_t^i | g_t^i, s_t)$ receives state $s_t$ and subgoal $g_t^i$ as input. The state $s_t$ consists of the humanoid's proprioceptive information, such as joint positions and velocities, and is shared across all policies. In contrast, the subgoal $g_t^i$ encodes task-specific information and varies depending on the task. The output of each policy, $a_t^i \in \mathbb{R}^d$, represents the target joint positions for the humanoid's entire body, which are then converted to torques using a proportional-derivative (PD) controller.

The low-level primitive policies are trained with dense rewards, incorporating task-relevant goal-matching rewards, regularization rewards, gait rewards, and other auxiliary objectives. While reward tuning is necessary to train these expert skills, the resulting policies are modular and reusable, allowing for seamless integration into high-level tasks with minimal additional reward engineering, and we anticipate that future humanoid manufacturers may directly provide such pretrained skills as standardized capabilities.

In this work, we focus on four broadly useful, task-agnostic primitive skills, though our approach can, in principle, accommodate an arbitrarily large skill library. Below, we provide a detailed description of these four primitive skills:

- **Walking**. The humanoid is required to walk robustly in response to a commanded velocity, enabling locomotion and basic mobility. The goal input consists of the desired velocity on the XY plane and yaw axis.
- **Reaching**. The humanoid remains stationary while reaching target 3D points in its surroundings using both wrists, supporting its manipulation capabilities. The goal input is the relative distance between the humanoid's wrist positions and their respective targets.
- **Squatting**. The humanoid squats down or stands up to reach a specified root height, facilitating adaptability to different workspaces. The goal input is the relative height between the humanoid's root and the target height.
- **Stepping**. The humanoid steps onto sampled ground points, enabling tasks involving foot-based, non-prehensile manipulation. Similar to **Reaching**, the goal input is the relative distance between the humanoid's feet and their respective targets on the floor.

### 3.3 Reusing and Blending Skills for High-Level Loco-Manipulation Tasks

Once the primitive skills are constructed, they can then be dynamically blended for novel tasks involving complex whole-body loco-manipulation, guided solely by task-specific rewards. In this blending process, all selected primitive skills are simultaneously activated, and their actions are weighted to accomplish challenging tasks beyond the capability of a single primitive policy. Unlike prior multi-expert approaches (Yang et al., 2020; Peng et al., 2019; Kumar et al., 2023), our approach proposed a unique vectorized weighting mechanism to blend different skills, enabling more flexible and accurate skill blending.

After the low-level primitive skills are constructed, given a high-level task specification, we first employ a skill selector to choose a subset of skills to blend. For example, a task requiring the humanoid to reach distant points would primarily rely on **Walking** and **Reaching**, while **Squatting** and **Stepping** would be less relevant, so only **Walking** and **Reaching** would be selected for blending. This selection process could be performed manually or by using foundation models leveraging their commonsense reasoning capabilities (Wei et al., 2022; Kuang et al., 2024b;a), as demonstrated in Fig. 5.

After selecting the relevant skills, we train a high-level controller $\pi_H$ that takes the current state and task-specific goal as input and outputs both the subgoals for the selected low-level goal-conditioned skills and the corresponding per-joint weight vectors used for blending their outputs. The final action is computed as a weighted sum of the actions from these low-level policies, as illustrated in Fig. 2.

Specifically, let the selected low-level primitive skills be $\{\pi_L^j, \ldots, \pi_L^k\}$, given the task goal $g_t$ and state $s_t$, $\pi_H$ network first outputs the raw subgoals $\{\tilde{g}_t^j, \ldots, \tilde{g}_t^k\}$ and the raw weight vectors $\{\tilde{W}_t^j, \ldots, \tilde{W}_t^k\}$, where each

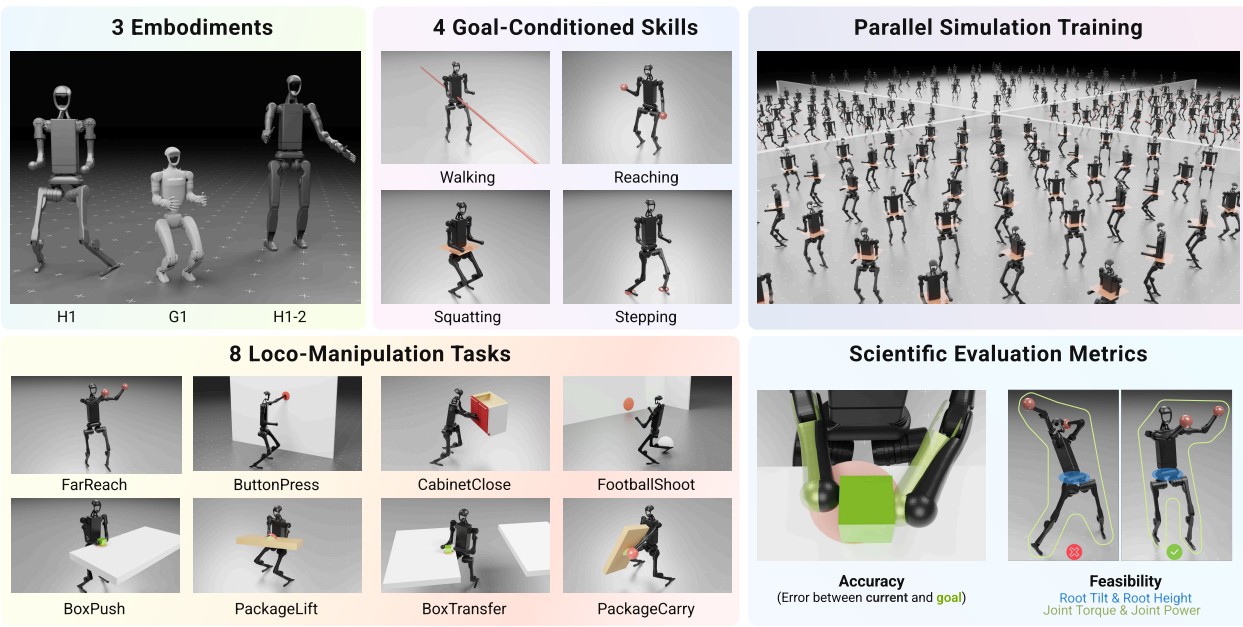

Figure 3: Our **SkillBench** is a parallel, cross-embodiment, and diverse simulated benchmark containing three embodiments, four primitive skills, and eight loco-manipulation tasks.

$\tilde{W}_t^i \in [0,1]^d$ is continuous and matches the dimensionality $d$ of the action space. The raw subgoals are then clamped to avoid exaggerated values, producing the processed subgoals $\{g_t^j, \dots, g_t^k\}$ as input for the low-level goal-conditioned skills.

Then, we add a Softmax layer to the raw weight vectors $\{\tilde{W}_t^j, \dots, \tilde{W}_t^k\}$ on the joint level as a layer of non-linearity in order to avoid direct linear combination that leads to reward hacking:

$$W_t^n[m] = \frac{e^{\tilde{W}_t^n[m]}}{\sum_{i=j}^{k} e^{\tilde{W}_t^i[m]}} \tag{1}$$

where $W_t^n[m]$ is the weight scalar of the $m$-th joint (element) of the $n$-th skill's weight vector. We verified the essentiality of the Softmax layer to provide non-linearity in Sec. 5.3.

After the high-level controller $\pi_H$ generates the final subgoals $\{g_t^j, \dots, g_t^k\}$ and per-joint weight vectors $\{W_t^j, \dots, W_t^k\}$, each low-level skill $\pi_L^i$ then concatenates its assigned subgoal $g_t^i$ with the current state $s_t$ as input to produce the action $a_t^i$. The final action $a_t$ is then computed by weighting all the actions as:

$$a_t = \sum_{i=j}^{k} a_t^i \odot W_t^i \tag{2}$$

where $\odot$ denotes the Hadamard (element-wise) product. During training, only the high-level controller is updated, while the low-level skills remain frozen. Notably, the blending process requires **only one or two** task-specific reward terms, significantly reducing the effort required for reward shaping.

## 4 SkillBench

To facilitate standardized humanoid learning research, we propose a new benchmark **SkillBench**, which is parallel, cross-embodiment, and diverse. We implement SkillBench in NVIDIA Isaac Gym (Makoviychuk et al., 2021) with PhysX physics engine, benefiting from its highly-parallelized simulation. SkillBench encompasses

3 distinct humanoid embodiments, 4 primitive skills, and 8 complex loco-manipulation tasks. In this section, we describe the details of SkillBench's simulated environments.

## 4.1 Observation and Action

We support various kinds of observations, which include (1) proprioception, including the humanoid's joint angles, joint velocities, last actions, base linear and angular velocities, and base Euler angles (projected gravity); (2) task-related states, such as the positions and rotations of objects in the scene; and (3) ego-centric vision, including RGB, depth (point clouds), and segmentation masks. In this work, we first consider state-based policies, using only proprioception and task-related states. We also include a preliminary investigation on vision-based RL in Sec. G.3.

Following previous works (He et al., 2024b;a; Fu et al., 2024; Zhang et al., 2024a), the action $a_t \in \mathbb{R}^d$ is the target joint positions of the whole-body joints on the humanoid, which is subsequently converted into torques using PD control.

## 4.2 Embodiments and Morphologies

As shown in Fig. 3, our SkillBench is designed for cross-embodiment compatibility, supporting three distinct humanoid models. We specifically choose Unitree humanoids due to their widespread adoption in recent years. In our SkillBench, we support the 19-DoF Unitree H1, 21-DoF Unitree G1, and 21-DoF Unitree H1-2. Their morphology details can be found in Sec. F.

In SkillBench, we fix all wrist and hand DoFs, as well as all torso DoFs except for the yaw axis, to simplify whole-body loco-manipulation. These DoFs can be enabled by users if needed. Under this configuration, H1 and H1-2 share similar overall sizes and shapes, while G1 and H1-2 share similar morphologies with 21 DoFs. This design aims to support future research on cross-embodiment humanoid learning by providing a standardized yet extensible platform.

## 4.3 Low-Level Skills and High-Level Tasks

Our SkillBench provides 4 primitive skills: `Walking`, `Reaching`, `Squatting`, and `Stepping`. These are generally applicable, broadly useful, and task-agnostic skills. For high-level tasks, we designed 8 complex loco-manipulation tasks that require whole-body coordination and are categorized into three difficulty levels: **Easy**, **Medium**, and **Hard**, based on task horizon and contact richness.

**Easy** tasks focus on short-horizon interactions with minimal contact:

- `FarReach`: Reach two distant 3D points using both hands.
- `ButtonPress`: Press a wall-mounted button with the left wrist while keeping the right arm's pose.
- `CabinetClose`: Close an open cabinet in front of the humanoid.

**Medium** tasks remain short-horizon but involve increased contact with objects and the environment:

- `FootballShoot`: Shoot a football towards a goal position.
- `BoxPush`: Push a box on a table to a target position.
- `PackageLift`: Lift a package to a specified height.

**Hard** tasks involve rich contact dynamics and require multi-stage, long-horizon coordination:

- `BoxTransfer`: Transfer a box from one table to a target location on another table.
- `PackageCarry`: Carry a package to a distant location.

Skill and task visualizations are shown in Fig. 3. Since G1 is shorter and smaller, objects and goal positions are scaled accordingly in G1 environments to ensure reachability. For further details, including task descriptions, task-specific goals, success checkers, reward functions, and additional parameters, please refer to Sec. D in the appendix.

| Task | FarReach | | | | | ButtonPress | | | | | CabinetClose | | | | |
|---|---|---|---|---|---|---|---|---|---|---|---|---|---|---|---|
| Metrics | **Error**↓ | Tilt↓ | h↑ | τ↓ | P↓ | **Error**↓ | Tilt↓ | h↑ | τ↓ | P↓ | **Error**↓ | Tilt↓ | h↑ | τ↓ | P↓ |
| PPO (Schulman et al., 2017) | **0.016±0.008** | 0.242 | 0.796 | 23.4 | 36.9 | 0.019±0.011 | 0.471 | 0.868 | 31.7 | 39.8 | **0.000±0.000** | 0.333 | 0.886 | 37.2 | 65.4 |
| DreamerV3 (Hafner et al., 2023) | 0.033±0.006 | 0.312 | 0.800 | 25.0 | 37.5 | 0.027±0.005 | 0.511 | 0.839 | 33.3 | 45.7 | 0.002±0.002 | 0.313 | 0.878 | 37.1 | 65.0 |
| HB (Sferrazza et al., 2024) | N/A | | | | | 0.347±0.178 | 0.453 | 0.877 | 30.7 | 93.7 | 0.051±0.218 | 0.097 | 0.802 | 22.7 | 29.4 |
| Sequential | 0.247±0.121 | 0.013 | 0.936 | 16.9 | 39.6 | 0.132±0.061 | 0.007 | 0.918 | 18.7 | 49.4 | 0.052±0.055 | 0.013 | 0.899 | 18.9 | 39.9 |
| MCP (Peng et al., 2019) | 0.045±0.031 | **0.018** | **0.969** | 14.1 | 20.5 | **0.005±0.003** | **0.016** | **0.910** | **13.9** | **19.2** | 0.001±0.004 | **0.061** | **0.916** | 15.0 | 21.0 |
| **Ours** | 0.021±0.012 | 0.045 | 0.955 | **13.5** | **20.2** | 0.009±0.007 | 0.041 | 0.848 | 16.8 | 20.3 | **0.000±0.000** | 0.119 | 0.903 | **13.6** | **16.0** |

Table 1: Quantitative comparison between our method and baseline methods on **H1-Easy** tasks.

| Task | FootballShoot | | | | | BoxPush | | | | | PackageLift | | | | |
|---|---|---|---|---|---|---|---|---|---|---|---|---|---|---|---|
| Metrics | **Error**↓ | Tilt↓ | h↑ | τ↓ | P↓ | **Error**↓ | Tilt↓ | h↑ | τ↓ | P↓ | **Error**↓ | Tilt↓ | h↑ | τ↓ | P↓ |
| PPO (Schulman et al., 2017) | 1.773±0.244 | 0.245 | 0.650 | 48.0 | 54.5 | 0.184±0.207 | 0.581 | 0.832 | 51.7 | 91.7 | 0.026±0.018 | 0.154 | 0.635 | 32.2 | 32.2 |
| DreamerV3 (Hafner et al., 2023) | 1.799±0.270 | 0.240 | 0.830 | 48.8 | 55.0 | 0.174±0.236 | 0.560 | 0.838 | 49.3 | 89.9 | 0.132±0.054 | 0.161 | 0.551 | 33.5 | 43.8 |
| HB (Sferrazza et al., 2024) | 1.684±0.187 | 0.093 | 0.849 | 23.8 | 45.4 | 0.125±0.039 | 0.119 | 0.803 | 17.9 | 14.9 | 0.571±0.193 | 1.143 | 0.561 | 39.2 | 68.2 |
| Sequential | 1.802±0.217 | 0.025 | 0.979 | 12.0 | 22.6 | 0.112±0.047 | 0.003 | 0.986 | 8.7 | 2.8 | 0.618±0.226 | 0.021 | 0.953 | 10.9 | 10.7 |
| MCP (Peng et al., 2019) | 1.604±0.263 | 0.054 | 0.888 | 20.0 | 42.8 | 0.037±0.039 | **0.020** | **0.884** | 12.6 | **5.9** | 0.485±0.116 | 0.032 | 0.832 | 13.7 | 9.1 |
| **Ours** | **1.109±0.285** | **0.131** | **0.843** | **26.1** | **92.8** | **0.009±0.007** | 0.064 | **0.884** | 15.0 | 9.9 | **0.024±0.069** | 0.062 | 0.717 | 21.0 | 15.1 |

Table 2: Quantitative comparison between our method and baseline methods on **H1-Medium** tasks.

## 4.4 Evaluation Metrics

As stated in Sec. 1, a scientific and comprehensive evaluation system is crucial for humanoid learning, rather than relying solely on task return comparisons. Therefore, our benchmark incorporates two types of metrics: the accuracy metric for assessing task success and a set of feasibility metrics to evaluate motion feasibility.

For the accuracy metric, we use **Error** (↓) to quantify the average deviation between the current and goal states. For example, in **FarReach**, **Error** is defined as the L1 distance between the humanoid's current wrist positions and the target positions. Across all tasks, **Error** is measured in meters (m). We also set a success threshold with regard to **Error** for each task. For feasibility metrics, we have:

- Tilt (↓), the average root pitch and roll angles, measured in radians (rad);
- Root Height $h$ (↑), measured in meters (m);
- Average Joint Torque $\tau$ (↓), measured in Newton-meters (N $\cdot$ m);
- Average Joint Power $P$ (↓), measured in Watts (W).

These feasibility metrics capture both stability (via Tilt and $h$) and action intensity (via $\tau$ and $P$), providing a comprehensive assessment of the overall feasibility of humanoid behaviors.

## 5 Experiments

### 5.1 Experimental Setup

On the H1 embodiment of SkillBench, we compare our method against two vanilla RL baselines learning from scratch: (1) model-free PPO (Schulman et al., 2017) and (2) model-based DreamerV3 (Hafner et al., 2023). We also compare against three hierarchical baselines: (1) the HumanoidBench baseline (HB) which uses a two-hand reaching policy as the low-level policy and then trains a task-specific high-level controller, (2) Sequential HRL which trains a high-level policy selector that decides which low-level skill to activate at each timestep and thereby sequence them, and (3) MCP (Peng et al., 2019) which leverages multiplicative compositional policies that synthesize scalar weights to average the low-level skills. We also compare our method with PPO (Schulman et al., 2017) on the G1 and H1-2 embodiments in Sec. G.1.

**All methods are trained with the same reward functions**, which are designed to be straightforward and incorporate **only one or two** intuitive task-specific terms (e.g., the distance between the current hand positions and target positions). We perform 20 rollouts per task for evaluation. To compute **Error**, we measure the state deviation between the final and goal states for each rollout, reporting both the mean and standard deviation. Feasibility metrics are first averaged over the duration of each rollout and subsequently averaged across all rollouts to assess motion feasibility throughout the task. Additional implementation details are provided in Sec. E.2.

| Task | BoxTransfer | | | | | PackageCarry | | | | |
|---|---|---|---|---|---|---|---|---|---|---|
| Metrics | **Error**↓ | Tilt↓ | h↑ | τ↓ | P↓ | **Error**↓ | Tilt↓ | h↑ | τ↓ | P↓ |
| PPO (Schulman et al., 2017) | 0.433±0.059 | 0.160 | 0.675 | 46.8 | 37.6 | 0.020±0.008 | 0.331 | 0.727 | 44.5 | 53.0 |
| DreamerV3 (Hafner et al., 2023) | 0.459±0.157 | 0.164 | 0.666 | 40.1 | 35.9 | 0.159±0.039 | 0.330 | 0.756 | 42.3 | 46.4 |
| HB (Sferrazza et al., 2024) | 0.459±0.047 | 0.131 | 0.762 | 25.6 | 23.0 | 0.443±0.120 | 0.336 | 0.838 | 18.3 | 21.7 |
| Sequential | 0.458±0.045 | 0.028 | 0.984 | 12.2 | 5.8 | 0.428±0.110 | 0.008 | 0.943 | 10.5 | 8.0 |
| MCP (Peng et al., 2019) | 0.421±0.026 | 0.034 | 0.894 | 15.3 | 7.8 | 0.152±0.030 | 0.032 | 0.871 | 18.0 | 28.4 |
| **Ours** | **0.007±0.004** | **0.055** | **0.884** | **16.7** | **17.0** | **0.013±0.008** | **0.043** | **0.787** | **21.3** | **28.9** |

Table 3: Quantitative comparison between our method and baseline methods on **H1-Hard** tasks.

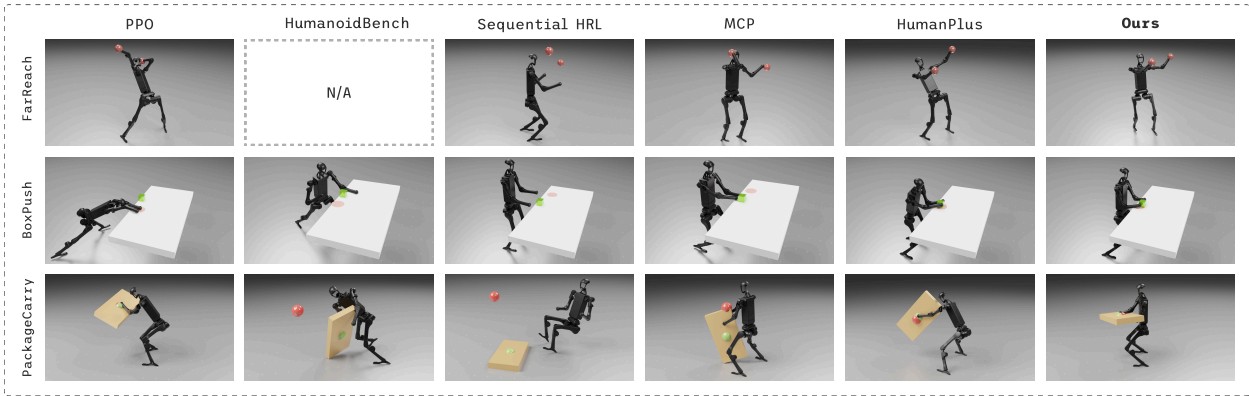

Figure 4: Qualitative comparison between different methods. Our **SkillBlender** not only achieves higher task accuracy, but also avoids reward hacking and yields more natural and feasible movements.

## 5.2 Results and Analysis

We show our main results of different difficulty levels in Table 1, 2, and 3. Cells highlighted in red indicate that the mean error exceeds the success threshold, signifying failure to learn a successful policy, while green denotes successful policies. We **bold** the best metric of all successful policies and underline the second-best. Our method significantly outperforms all baselines across most tasks and metrics, highlighting its clear advantages with respect to task success and motion feasibility. Qualitative comparisons in Fig. 4 further show that our method produces more accurate, natural, and feasible behaviors compared to baseline approaches.

We first compare our framework with vanilla RL algorithms that learn these tasks from scratch. Although vanilla PPO (Schulman et al., 2017) and DreamerV3 (Hafner et al., 2023) can succeed in **Easy** tasks, they struggle with most **Medium** and **Hard** tasks. Moreover, vanilla RL exhibits severe reward hacking on these easy tasks with extreme motions, as shown in their poor feasibility metrics and qualitative results. In contrast, our method shows more accurate, natural, and feasible behaviors on all tasks due to our more structured exploration and more flexible skill blending.

We also compare our method with ones with different hierarchy designs. Compared to the HumanoidBench baseline (HB) (Sferrazza et al., 2024), our method shows consistently better performance, due to our better low-level representations that not only decouple different humanoid functionalities but also provide natural regularization that mitigates reward hacking. Compared with Sequential HRL adopted in many previous works (Li et al., 2020; Ji et al., 2022; Cheng et al., 2023), we found that in humanoid learning, this paradigm leads to far worse performance than skill blending, failing to learn all the possible tasks. This occurs because, for humanoids, each primitive skill controls a specific body range, making it more effective to activate multiple skills simultaneously for whole-body coordination. For example, when carrying a box, both `Walking` and `Reaching` should be activated at the same time to hold the box while moving. Compared to MCP (Peng et al., 2019) which has comparable or slightly better performance to ours on certain easy tasks, it struggles with harder tasks, barely successfully learning them except `BoxPush`, emphasizing the superiority of our vectorized skill blending mechanism that fosters learning complex tasks.

Our analysis suggests that the strength of our framework stems from the structural priors from the low-level primitive skills that provide extra robustness and regularization, effectively reducing the RL search space, improving sample efficiency, and mitigating reward hacking. Our method simplifies the problem of task-specific

| Task | FarReach | | | | | BoxPush | | | | | PackageCarry | | | | |
|---|---|---|---|---|---|---|---|---|---|---|---|---|---|---|---|
| Metrics | **Error**↓ | Tilt↓ | $h$↑ | $\tau$↓ | $P$↓ | **Error**↓ | Tilt↓ | $h$↑ | $\tau$↓ | $P$↓ | **Error**↓ | Tilt↓ | $h$↑ | $\tau$↓ | $P$↓ |
| w/o `Walking` | 0.408±0.223 | 0.051 | 0.835 | 16.6 | 9.3 | 0.032±0.028 | **0.021** | 0.874 | 16.8 | 10.1 | 0.383±0.100 | 0.030 | 0.628 | 18.8 | 14.4 |
| w/o `Reaching` | 0.172±0.061 | 0.039 | 0.995 | 12.9 | 27.7 | 0.065±0.093 | 0.039 | 0.932 | 13.2 | 20.5 | 0.362±0.077 | 0.033 | 0.836 | 12.8 | 6.1 |
| w/o Softmax | 0.032±0.023 | 0.129 | 0.821 | 22.3 | 29.4 | 0.094±0.048 | 0.209 | 0.796 | 50.6 | 44.9 | 0.046±0.018 | 0.163 | 0.603 | 39.9 | 50.3 |
| HumanPlus (Fu et al., 2024) | 0.024±0.008 | 0.207 | 0.915 | 19.1 | 34.2 | 0.015±0.008 | 0.228 | **0.884** | 21.5 | 14.3 | 0.023±0.014 | 0.258 | 0.742 | 27.3 | 35.6 |
| ExBody (Cheng et al., 2024) | 0.049±0.030 | 0.051 | **0.980** | 13.8 | 20.8 | 0.021±0.013 | 0.036 | 0.877 | 16.2 | 11.8 | 0.413±0.072 | 0.093 | 0.784 | 13.0 | 9.4 |
| **Ours** | **0.021±0.012** | 0.045 | 0.955 | 13.5 | 20.2 | **0.009±0.007** | 0.064 | **0.884** | 15.0 | 9.9 | **0.013±0.008** | 0.043 | 0.787 | 21.3 | 28.9 |

Table 4: Ablation studies on tasks of different difficulty levels. Our method shows consistently better performance, validating the effectiveness of our framework design.

policy optimization by constraining the search space to combinations of high-quality, pretrained primitives, which inherently regularizes behaviors to minimize reward hacking. As such, SkillBlender not only significantly outperforms all baselines in task success but also yields more feasible and well-behaved humanoid motion.

## 5.3 Ablation Studies

To further analyze our framework design, we conduct ablation studies on various components to highlight the importance of each element in our method and validate different hierarchy designs. We evaluate on three tasks—**FarReach**, **BoxPush**, and **PackageCarry**—representing different difficulty levels, using the H1 robot. The results for the task **Error** across different ablation methods are shown in Table 4. As illustrated, removing either **Walking** or **Reaching** leads to significant performance degradation due to a restricted search space, highlighting the essential role of basic primitive skills.

We also verified the effectiveness of the Softmax layer on the raw weights produced by the high-level controller network, as in Eq. 1. As shown in the results, removing the Softmax layer leads to significantly worse performance, especially in feasibility metrics. This is in line with the non-linearity provided by the Softmax layer, which effectively produces weight constraints, reduces reward hacking, and generates more natural and feasible movements.

We also experimented with using human motion trackers as low-level policies (Hansen et al., 2024). We train HumanPlus (Fu et al., 2024) and ExBody (Cheng et al., 2024) using the aligned state $s_t$ same as our settings, and train a high-level controller to output the tracker's input goal for each task. As shown in Table 4, both trackers underperform our SkillBlender, demonstrating the superiority of our skill blending hierarchy. HumanPlus (Fu et al., 2024) tracks whole-body motions, which can be seen as a regularized version of PPO. However, this regularization also leads to the policy's inability to achieve higher task success. Moreover, it does not fully resolve the reward hacking issue as shown in Fig. 4. In ExBody (Cheng et al., 2024), humanoid control is split into two body parts, limiting the high-level controller's exploration, making it difficult to learn complex tasks. In contrast, our method is more accurate, thanks to its more structured and versatile action space derived from different primitive skills and their dynamic blending.

## 6 Conclusions, Limitations, and Future Works

**Conclusions** In this paper, we introduced **SkillBlender**, a novel pretrain-then-blend framework for versatile humanoid whole-body loco-manipulation. At the core of SkillBlender is to pretrain primitive skills and dynamically blend them for complex loco-manipulation tasks with minimal reward engineering. We also proposed a new benchmark, **SkillBench**, which is parallel, cross-embodiment, and diverse, to benchmark humanoid whole-body loco-manipulation scientifically. Extensive simulated experiments demonstrate the effectiveness of our framework, showcasing SkillBlender's capabilities of performing complex and challenging whole-body loco-manipulation tasks accurately and naturally. We hope our method and benchmark can benefit future open research on humanoid learning.

**Limitations and Future Works** Despite compelling results, our work has certain limitations that can be further improved in future works. First, our study primarily focuses on whole-body loco-manipulation using the humanoid's forearms, without incorporating specific end-effectors such as parallel grippers or dexterous hands. Additionally, we have not yet deployed our high-level task policies in real-world scenarios, due to the reliance on state-based observations and the sim2real gap. As part of future work, we plan to explore more

effective real2sim physics alignment techniques (e.g., (He et al., 2025a)) to enable agile and robust humanoid movements in the real world. We also hope that future research in related domains — such as design and development of simulation-friendly humanoid hardware and advanced vision-based reinforcement learning algorithms — will help address these challenges and further advance the field of humanoid learning.

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

# A  Observation Space

For state-based policies, the observation space for the actor (goal $g_t$ and state $s_t$) comprises $3d + 6 + N$ dimensions ($d$ is the robot's DoF), in which $s_t$ comprises $3d + 6$ dimensions (joint angles, joint velocities, last actions, base angular velocity, and base Euler angles or projected gravity), and $g_t$ comprises of $N$ dimensions based on the task specification. Observation space for vision-based policies is discussed in Sec. G.3.

# B  Action Space

The action space for each low-level primitive skill $\pi_L^i$ (or baseline methods like PPO (Schulman et al., 2017), DreamerV3 (Hafner et al., 2023)) is $\mathbb{R}^d$. And for the high-level controller $\pi_H$ that blends low-level primitive skills $\{\pi_L^j, \ldots, \pi_L^k\}$, the action dimension $d_H$ is:

$$d_H = \sum_{i=j}^{k} (N_i + d) \tag{3}$$

comprising the subgoal $g_t^i \in \mathbb{R}^{N_i}$ and vector weight $W_t^i \in [0, 1]^d$ for each primitive skill $\pi_L^i$.

# C  Low-Level Primitive Skill Specifications

## C.1  `Walking`

**Objective.** Walk in a given velocity command.

**Goal input.** Linear velocity command on xy axes and angular velocity command on the yaw axis.

## C.2  `Reaching`

**Objective.** Reach two 3D target points using its two wrists while standing still.

**Goal input.** The relative distances between the humanoid's wrists and respective target points.

## C.3  `Squatting`

**Objective.** Squat down or stand up to reach a target root height.

**Goal input.** The relative root height between the current root height and the target height.

## C.4  `Stepping`

**Objective.** Step its two feet on sampled points on the ground.

**Goal input.** The relative distances between the humanoid's feet and respective target points.

# D  High-Level Loco-Manipulation Task Specifications

In this section, we take Unitree H1 as a representative platform to describe our high-level task specifications, including the task objective, goal input, skills, success threshold, and reward function. The reward functions are kept simple and intuitive, consisting of **only one or two** terms, thereby requiring **minimal reward engineering**.

## D.1  `FarReach`

**Objective.** Reach two distant 3D points using both hands.

**Goal input.** The relative distances between the humanoid's wrists and respective target points.

**Skills. Walking + Reaching**

**Success threshold.** $0.05m$.

**Reward.** The reward function is defined as:

$$R(s, a) = 5e^{-4\|p_{wr} - \hat{p_{wr}}\|} \tag{4}$$

### D.2 ButtonPress

**Objective.** Press a wall-mounted button with the left wrist while keeping the right arm's pose.

**Goal input.** The relative distances between the humanoid's left wrist and the button.

**Skills. Walking + Reaching**

**Success threshold.** $0.05m$.

**Reward.** The reward function is defined as:

$$R(s, a) = 5e^{-4\|p_{wr} - p_{bn}\|} + 0.5e^{-4\|q_{ra}\|} \tag{5}$$

### D.3 CabinetClose

**Objective.** Close an open cabinet in front of the humanoid.

**Goal input.** (1) The articulation angles for the articulated cabinet's two doors; (2) The relative distances between the humanoid's wrists and the cabinet.

**Skills. Walking + Reaching**

**Success threshold.** $0.01m$.

**Reward.** The reward function is defined as:

$$R(s, a) = 5e^{-4\|p_{wr} - p_{arti}\|} + 5e^{-4\|q_{arti}\|} \tag{6}$$

### D.4 FootballShoot

**Objective.** Shoot a football towards a goal position.

**Goal input.** (1) The relative distance between the ball and the goal; (2) The relative distance between the humanoid's torso and the ball.

**Skills. Walking + Stepping**

**Success threshold.** $1.5m$.

**Reward.** The reward function is defined as:

$$R(s, a) = e^{-4\|p_{torso}^{xy} - p_{oriball}^{xy}\|} + 5e^{-\|p_{ball} - p_{goal}\|} \tag{7}$$

### D.5 BoxPush

**Objective.** Push a box on a table to a target position.

**Goal input.** (1) The relative distance between the box and the target; (2) The relative distance between the humanoid's wrists and the box.

**Skills. Walking + Reaching**

**Success threshold.** $0.05m.$

**Reward.** The reward function is defined as:

$$R(s, a) = 5e^{-4\|p_{box} - \hat{p_{box}}\|} + 5e^{-4\|p_{wr} - p_{box}\|} \tag{8}$$

### D.6 `PackageLift`

**Objective.** Lift a package to a specified height.

**Goal input.** (1) The relative distance between the package and the target; (2) The relative distance between the humanoid's wrists and the package.

**Skills. Reaching + Squatting**

**Success threshold.** $0.1m.$

**Reward.** The reward function is defined as:

$$R(s, a) = 5e^{-4\|h_{pkg} - \hat{h_{pkg}}\|} + 5e^{-4\|p_{wr} - p_{pkg}\|} \tag{9}$$

### D.7 `BoxTransfer`

**Objective.** Transfer a box from one table to a target location on another table.

**Goal input.** (1) The relative distance between the box and the target; (2) The relative distance between the humanoid's wrists and the box.

**Skills. Walking + Reaching**

**Success threshold.** $0.05m.$

**Reward.** The reward function is defined as:

$$R(s, a) = 5e^{-4\|p_{box} - \hat{p_{box}}\|} + e^{-4\|p_{wr} - p_{box}\|} \tag{10}$$

### D.8 `PackageCarry`

**Objective.** Carry a package to a distant location.

**Goal input.** (1) The relative distance between the package and the target; (2) The relative distance between the humanoid's wrists and the package.

**Skills. Walking + Reaching + Squatting**

**Success threshold.** $0.1m.$

**Reward.** The reward function is defined as:

$$R(s, a) = 5e^{-4\|p_{pkg} - \hat{p_{pkg}}\|} + 5e^{-4\|p_{wr} - p_{pkg}\|} \tag{11}$$

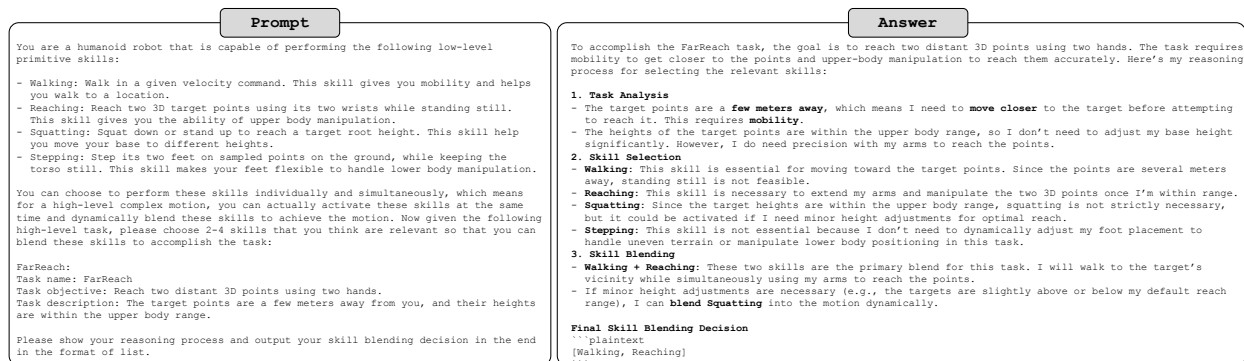

Figure 5: An example of GPT-4o reasoning to perform skill selection on the `FarReach` task.

# E    Implementation Details

## E.1    Skill Selector

In this work, the skills are manually selected to blend. However, given the task and skill specifications and descriptions, we can leverage an LLM to perform common-sense reasoning (Wei et al., 2022) to select relevant skills automatically. In Fig. 5 we show an example of GPT-4o selecting skills for the `FarReach` task.

## E.2    Baseline Implementations

We use the standard PPO (Schulman et al., 2017) and DreamerV3 (Hafner et al., 2023) implementations for our baselines. And for HumanoidBench baseline (HB) (Sferrazza et al., 2024), we freeze the PPO-trained `FarReach` policy as the low-level two-hand reaching policy and then train a high-level controller, which is also implemented in PPO. Note that for HB, `FarReach` is not applicable (N/A) since there is no high-level controller.

For all model-free RL methods, we use 4096 parallel environments during training. Due to training stability and speed constraints, we set the number of parallel environments to 16 for the model-based DreamerV3 (Hafner et al., 2023) baseline.

## E.3    Neural Network Architechtures

All state-based policy networks are implemented as end-to-end MLPs. Actors are all MLPs with $[512, 256, 128]$ hidden units, and critics are all MLPs with $[768, 256, 128]$ hidden units. Critics can also access privileged observations such as base linear velocity, body mass, contact mask, etc. For visual RL, the image encoders are implemented as vanilla CNNs.

## E.4    Training Details

For all goal-conditioned model-free RL methods in this work, we employ Proximal Policy Optimization (PPO) (Schulman et al., 2017) to optimize the policy. For PPO training, we set the entropy coefficient to 0.001, learning rate to `1e-5`, and number of mini-batches to 4. In addition, we put $\gamma = 0.994$ and $\lambda = 0.9$.

## E.5    Compute Resources

We use NVIDIA RTX 4090/A6000/A100 GPU for training our low-level skills and high-level controllers. All the training and inference are done on a single GPU. RAM required is less than 24GB in the training stage, and less than 12GB in the inference stage. For easy tasks like `FarReach` or `ButtonPress` it typically takes 12-24 hours to finish training, while harder tasks take longer, typically 24-72 hours.

| Task | FarReach | | | | | ButtonPress | | | | | CabinetClose | | | | |
|---|---|---|---|---|---|---|---|---|---|---|---|---|---|---|---|
| Metrics | **Error**↓ | Tilt↓ | $h$↑ | $\tau$↓ | $P$↓ | **Error**↓ | Tilt↓ | $h$↑ | $\tau$↓ | $P$↓ | **Error**↓ | Tilt↓ | $h$↑ | $\tau$↓ | $P$↓ |
| PPO (Schulman et al., 2017) | **0.019±0.009** | 0.220 | **0.754** | 13.3 | 32.5 | **0.014±0.007** | 0.930 | 0.573 | 20.9 | 56.5 | 0.622±0.469 | 0.552 | 0.674 | 25.8 | 63.6 |
| **Ours** | 0.023±0.018 | **0.214** | 0.715 | **11.0** | **30.4** | 0.032±0.083 | **0.196** | **0.647** | **13.2** | **38.2** | **0.000±0.000** | **0.234** | 0.647 | **12.6** | **22.6** |

Table 5: Quantitative comparison between our method and baseline methods on **G1-Easy** tasks.

| Task | FootballShoot | | | | | BoxPush | | | | | PackageLift | | | | |
|---|---|---|---|---|---|---|---|---|---|---|---|---|---|---|---|
| Metrics | **Error**↓ | Tilt↓ | $h$↑ | $\tau$↓ | $P$↓ | **Error**↓ | Tilt↓ | $h$↑ | $\tau$↓ | $P$↓ | **Error**↓ | Tilt↓ | $h$↑ | $\tau$↓ | $P$↓ |
| PPO (Schulman et al., 2017) | 1.733±0.236 | 0.309 | 0.756 | 20.8 | 79.6 | 0.075±0.128 | 0.462 | 0.657 | 25.1 | 65.3 | 0.226±0.070 | 0.545 | 0.557 | 25.8 | 36.3 |
| **Ours** | **1.476±0.276** | **0.190** | **0.664** | **17.7** | **70.4** | **0.039±0.096** | **0.176** | **0.615** | **15.4** | **54.3** | **0.074±0.078** | **0.346** | **0.773** | **20.4** | **57.6** |

Table 6: Quantitative comparison between our method and baseline methods on **G1-Medium** tasks.

| Task | BoxTransfer | | | | | PackageCarry | | | | |
|---|---|---|---|---|---|---|---|---|---|---|
| Metrics | **Error**↓ | Tilt↓ | $h$↑ | $\tau$↓ | $P$↓ | **Error**↓ | Tilt↓ | $h$↑ | $\tau$↓ | $P$↓ |
| PPO (Schulman et al., 2017) | 0.489±0.047 | 0.538 | 0.676 | 26.7 | 37.5 | 0.291±0.055 | 1.088 | 0.313 | 29.7 | 102.5 |
| **Ours** | 0.340±0.021 | 0.304 | 0.496 | 15.9 | 50.9 | 0.058±0.069 | 0.176 | 0.324 | 15.0 | 47.4 |

Table 7: Quantitative comparison between our method and baseline methods on **G1-Hard** tasks.

| Task | FarReach | | | | | ButtonPress | | | | | CabinetClose | | | | |
|---|---|---|---|---|---|---|---|---|---|---|---|---|---|---|---|
| Metrics | **Error**↓ | Tilt↓ | $h$↑ | $\tau$↓ | $P$↓ | **Error**↓ | Tilt↓ | $h$↑ | $\tau$↓ | $P$↓ | **Error**↓ | Tilt↓ | $h$↑ | $\tau$↓ | $P$↓ |
| PPO (Schulman et al., 2017) | **0.013±0.005** | 0.309 | 0.914 | 32.0 | 37.6 | 0.027±0.017 | 0.392 | 0.728 | 26.9 | 31.2 | 0.001±0.003 | 0.339 | 0.779 | 45.7 | 53.7 |
| **Ours** | 0.049±0.021 | **0.077** | **0.919** | **15.2** | **18.5** | **0.023±0.024** | **0.090** | **0.879** | **18.9** | **26.5** | **0.000±0.000** | **0.121** | **0.897** | **20.0** | **24.6** |

Table 8: Quantitative comparison between our method and baseline methods on **H1-2-Easy** tasks.

| Task | FootballShoot | | | | | BoxPush | | | | | PackageLift | | | | |
|---|---|---|---|---|---|---|---|---|---|---|---|---|---|---|---|
| Metrics | **Error**↓ | Tilt↓ | $h$↑ | $\tau$↓ | $P$↓ | **Error**↓ | Tilt↓ | $h$↑ | $\tau$↓ | $P$↓ | **Error**↓ | Tilt↓ | $h$↑ | $\tau$↓ | $P$↓ |
| PPO (Schulman et al., 2017) | 1.569±0.244 | 0.232 | 0.796 | 31.3 | 129.2 | **0.040±0.027** | 0.311 | 0.862 | 38.6 | 34.1 | 0.470±0.118 | 1.272 | 0.434 | 48.9 | 66.9 |
| **Ours** | **1.400±0.186** | **0.095** | **0.871** | **27.0** | **80.4** | 0.050±0.094 | **0.087** | **0.884** | **17.1** | **11.6** | 0.557±0.189 | 0.079 | 0.686 | 16.3 | 18.9 |

Table 9: Quantitative comparison between our method and baseline methods on **H1-2-Medium** tasks.

| Task | BoxTransfer | | | | | PackageCarry | | | | |
|---|---|---|---|---|---|---|---|---|---|---|
| Metrics | **Error**↓ | Tilt↓ | $h$↑ | $\tau$↓ | $P$↓ | **Error**↓ | Tilt↓ | $h$↑ | $\tau$↓ | $P$↓ |
| PPO (Schulman et al., 2017) | 0.397±0.024 | 0.393 | 0.962 | 32.0 | 27.9 | 0.373±0.061 | 0.802 | 0.594 | 35.8 | 43.8 |
| **Ours** | 0.315±0.046 | 0.072 | 0.829 | 17.4 | 13.6 | **0.051±0.019** | 0.038 | 0.732 | 23.2 | 34.3 |

Table 10: Quantitative comparison between our method and baseline methods on **H1-2-Hard** tasks.

# F  Morphology Details

As shown in Fig. 3, our SkillBench is designed for cross-embodiment compatibility, supporting three distinct humanoid models. We specifically choose Unitree humanoids due to their widespread adoption in recent years. The supported models include:

- **Unitree H1.** The most widely used humanoid embodiment in prior works (He et al., 2024b;a; Fu et al., 2024; Zhang et al., 2024a). It stands approximately 1.7 meters tall and features 19 degrees of freedom (DoFs), including two 3-DoF shoulders, two 1-DoF elbows, a 1-DoF yaw joint in the torso, two 3-DoF hips, two 1-DoF knees, and two 1-DoF pitch ankle joints.
- **Unitree G1.** A smaller humanoid model, standing around 1.2 meters tall. It features an additional roll DoF on each ankle, increasing the total DoF count to 21.
- **Unitree H1-2.** Morphologically similar to G1, with 2-DoF ankles and a total of 21 DoFs, but comparable in size and shape to H1.

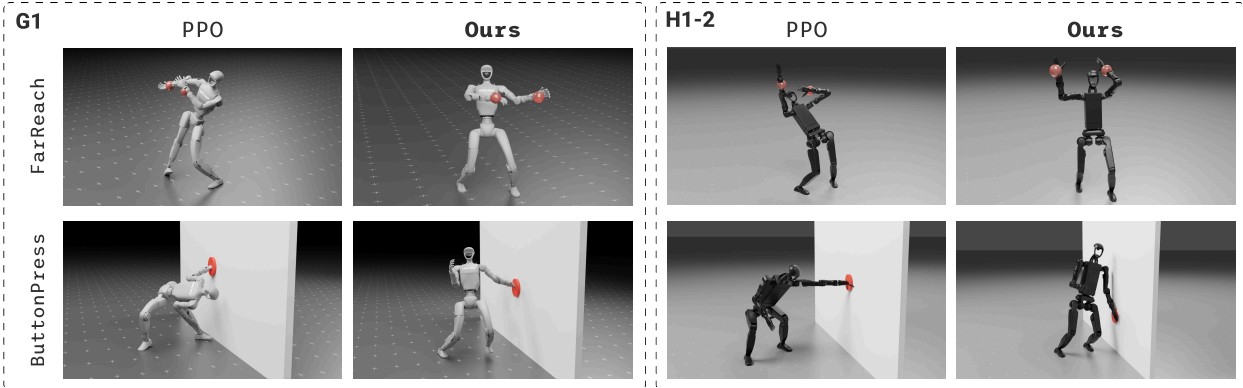

Figure 6: Qualitative results on G1 and H1-2 embodiments. Our method produces more accurate and natural movements, validating our framework's superiority across multiple embodiments.

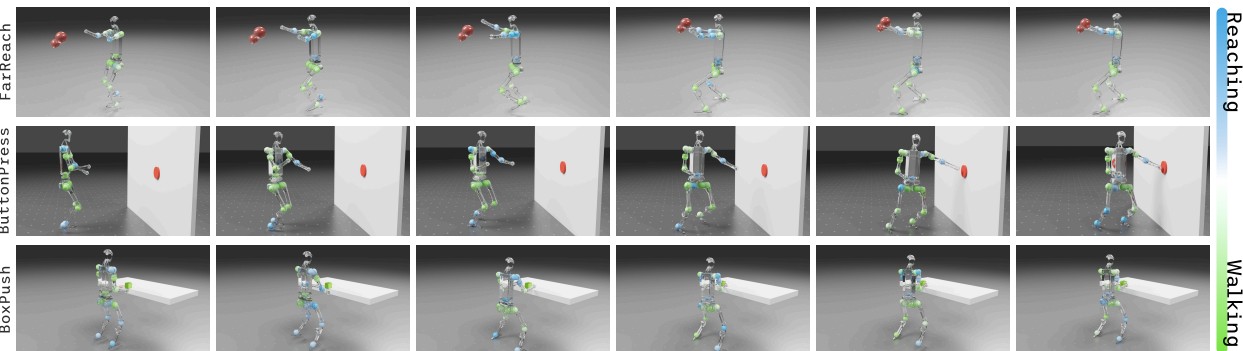

Figure 7: Visualization of whole-body per-joint weights at different stages of three different tasks. More blue means more `Reaching`, and more green means more `Walking`.

# G    Additional Experiments

## G.1    Results on G1 and H1-2 Embodiments

We also analyze the performance of different humanoid embodiments across various tasks. We compare our method with PPO (Schulman et al., 2017) on the G1 and H1-2 embodiments. Quantitative results are shown in Table. 5, 6, 7, and 8, 9, 10, and we illustrate qualitative results in Fig. 6. As shown in the results, our method outperforms PPO by a large margin, producing more accurate and natural movements, demonstrating our framework's superiority across multiple humanoid embodiments. Comparing H1 results with G1 and H1-2, we also observe that G1 and H1-2 generally exhibit higher task errors than H1. This discrepancy is primarily due to their increased degrees of freedom (DoFs), particularly the additional ankle roll DoFs, which introduce greater instability. These results suggest that even a small increase in foot articulation can significantly increase task difficulty.

## G.2    Skill Blending Decomposition

To provide a more intuitive understanding of our framework, especially our proposed skill blending mechanism, Fig. 7 visualizes the whole-body per-joint weights at different stages of **FarReach**, **ButtonPress**, and **BoxPush**, all of which blend **Walking** and **Reaching**. This visualization highlights the spatial-temporal decomposition of our skill blending, where the two skills interleave rather than one skill dominating the overall motion. For example, in **FarReach**, we observe a clear spatial decomposition: **Walking** primarily influences the lower body, while **Reaching** governs upper-body movements. Similarly, in **ButtonPress**, the contribution of **Reaching** progressively increases throughout the task, particularly as the left wrist approaches the button.

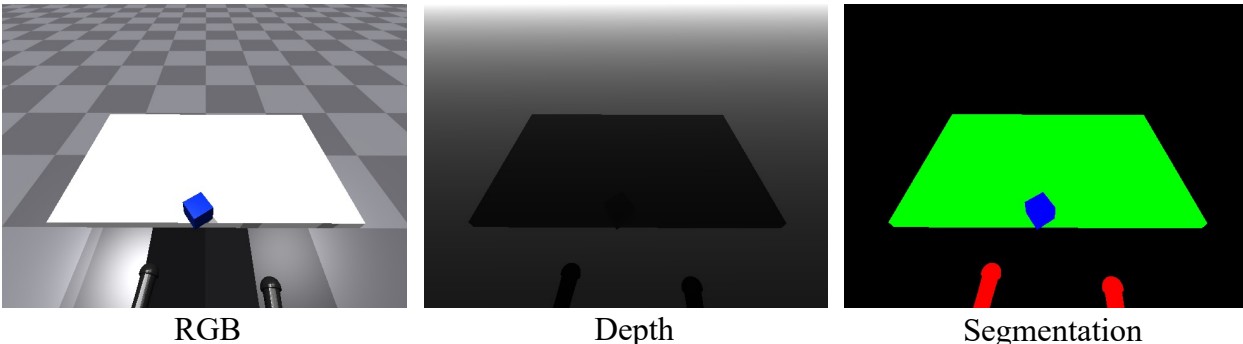

| RGB | Depth | Segmentation |

Figure 8: Ego-centric visual observations, including RGB, depth (point cloud), and segmentation masks.

| # Environments | 4 | 64 |
|---|---|---|
| PPO (Schulman et al., 2017) | 21.53 | 32.34 |
| DreamerV3 (Hafner et al., 2023) | 25.42 | 34.00 |
| **Ours** | **28.14** | **37.90** |

Table 11: Maximum task mean reward ($\uparrow$) on **vision-based** H1 `BoxPush` task.

### G.3 Vision-Based Policy Learning

Although our framework **SkillBlender** is currently state-based, we have also conducted some preliminary research on vision-based RL, given its high potential in real-world applications. As shown in Fig. 8, our **SkillBench** supports ego-centric visual observations, including RGB images, depth images (point clouds), and segmentation masks. In this work, we trained vision-based policies using ego-centric RGB images, using PPO (Schulman et al., 2017), DreamerV3 (Hafner et al., 2023), and our **SkillBlender**, on the H1 `BoxPush` task. Due to the rendering and training speed, we set the number of parallel environments to 4 and 64, and the image resolution to $64 \times 48$.

In Table 11, we present the maximum task mean reward for different methods across various parallel environment settings. Our method outperforms PPO and DreamerV3 in both the 4 and 64 environment configurations, demonstrating its effectiveness in both state-based and vision-based settings. However, it is important to note that due to the challenges in visual RL and limited parallel environments, none of these methods were able to successfully complete the task, which is why we focus on comparing task mean rewards. This highlights the importance of highly parallelized simulations. We hope that future advancements in efficient and high-quality parallel rendering will further support ego-centric vision-based humanoid learning.

### G.4 Additional Ablations

We present further ablation results in Table 12, including experiments and analysis on our skill selector, high-level control frequency, and Softmax layer. We conduct experiments on the H1 `BoxPush` task — which blends `Walking` and `Reaching` — and compare with several variants, including (1) removing the skill selector and blending all four skills, (2) repeating selected skills twice, (3) downgrading the output frequency of $\pi_H$ from 100Hz to 10Hz, and (4) replacing the Softmax layer with an alternative non-linear layer (L2 normalization).

As shown in the results, while the performance of our method without the skill selector slightly drops compared to the full model, it could still successfully learn the high-level task and achieve decent behaviors. This indicates that our proposed vectorized skill blending paradigm is robust to the inclusion of irrelevant skills, and our proposed skill selector could further help improve efficiency and performance by reducing redundancy. This supports the scalability of our framework to broader skill sets, as more skills are introduced over time.

| Metrics | **Error**↓ | Tilt↓ | $h$↑ | $\tau$↓ | $P$↓ |
|---|---|---|---|---|---|
| w/o Skill Selector | 0.030±0.021 | 0.097 | 0.890 | 19.7 | 11.6 |
| w/ Repeated Skills | 0.015±0.008 | 0.074 | **0.899** | **13.9** | **9.7** |
| 10Hz $\pi_H$ | 0.012±0.006 | 0.070 | 0.901 | 14.6 | 19.7 |
| L2 Normalization | 0.095±0.048 | 0.163 | 0.849 | 15.8 | 16.6 |
| **Ours** | **0.009±0.007** | **0.064** | 0.884 | 15.0 | 9.9 |

Table 12: Additional ablation studies on H1 `BoxPush` task.

Moreover, by repeating relevant skills, the overall performance remains largely comparable, yielding similar results and behaviors. This further reinforces our finding that our proposed skill blending framework remains robust even as more skills — whether relevant or not — are included.

We also find that downgrading the high-level control frequency from 100Hz to 10Hz will slightly affect performance, but a 100Hz frequency could allow the high-level controller to be fully closed-loop and highly responsive to changes in the environment and ensure maximum reactivity to external disturbances or task dynamics. Using a persistent sub-goal over multiple steps can be viewed as a special case of our setup with a lower update frequency, but it introduces a more open-loop behavior, potentially reducing adaptability in dynamic scenarios.

Finally, to further evaluate the necessity of the Softmax layer, we replace our joint-level Softmax with joint-level L2 normalization. The results show that the L2-normalized variant struggled to learn meaningful behavior, completely failing to blend skills effectively. We attribute this to the fact that L2 normalization only controls the magnitude of the vector but not its structure — it lacks the inductive bias that Softmax imposes, leading to a looser, less structural action space.

## H Real-World Skill Deployment

We utilize a Unitree H1 humanoid robot to deploy our simulation-trained policies in the real world as a sanity check on sim2real transfer. As shown in Fig. 9, we successfully deployed our primitive skills in the real world and controlled them with goal conditions. For robust sim2real transfer, we leverage larger domain randomization and incorporate projected gravity input. Video results can be found in the supplementary material, where we control the primitive skills to perform various task-agnostic periodical movements. In future works, we aim to distill our state-based high-level task policies into vision-based policies and directly deploy them to the real world.

## I Common Failure Cases

We observe several common failure modes in both our method and baseline methods. For primitive skills, despite applying domain randomization during simulation training, policies can struggle when faced with out-of-distribution states — such as highly unusual initial poses — leading to failure to initiate motion properly. This issue is further exacerbated in real-world settings, where sensor noise is more significant. For high-level tasks in simulation, particularly **Hard** tasks with longer horizons (e.g. `BoxTransfer`), we occasionally observe that the humanoid prematurely ceases exploration upon receiving a relatively high intermediate reward. This results in suboptimal behaviors where the humanoid stays in local minima without completing the full task.

## J Broader Impacts

This work has the potential for several positive societal impacts, including applications in elder care, assistance in hazardous or inaccessible environments, and improved autonomy in service robotics. However, potential negative consequences include job displacement and increased overreliance on robotic systems. We encourage

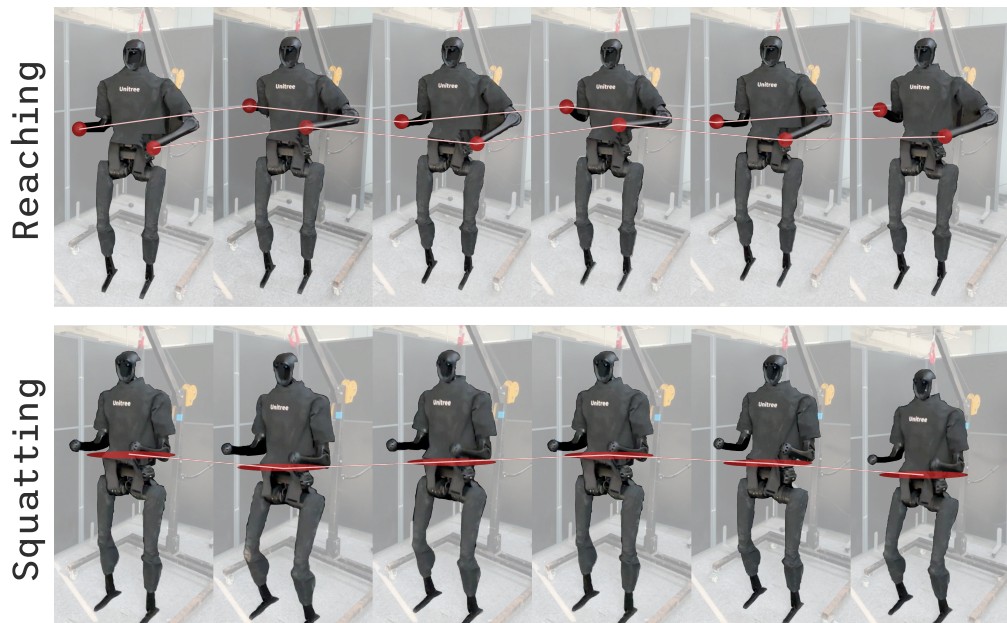

Figure 9: Demonstrations of our primitive skill sim2real deployment. We control the humanoid to perform periodical **Reaching** and **Squatting**.

responsible development and deployment of such technologies and hope our contributions ultimately lead to broad societal benefit.

