# OpenReview forum: "SkillBlender: Towards Versatile Humanoid Whole-Body Loco-Manipulation via Skill Blending"
_TMLR — Rejected by TMLR_

### Review · Reviewer_o6cz · 2025-11-10

**Summary Of Contributions:**

The paper tackles the challenging problem of versatile humanoid whole-body loco-manipulation, which is crucial for deploying humanoid robots in daily environments. Its originality lies in SkillBlender, a pretrain-then-blend hierarchical framework that combines goal-conditioned primitive skills with a high-level controller using a vectorized weighting mechanism, inspired by human motor skill acquisition. The proposed SkillBench benchmark provides a parallel, cross-embodiment, and diverse evaluation platform with metrics assessing both task accuracy and motion feasibility, supporting systematic and reproducible research. Overall, the work demonstrates solid technical quality, a well-structured hierarchical approach, and potential to advance scalable and versatile humanoid learning.

**Audience:**

Yes

**Audience Explanation:**

A novel hierarchical reinforcement learning framework for versatile humanoid loco-manipulation is very important

**Claims And Evidence:**

No

**Claims Explanation:**

Some statement is not very convincing, please see as the "Request Changes"

**Requested Changes:**

1. Clarity and Scope
   The introduction clearly motivates the challenges of versatile humanoid whole-body loco-manipulation and provides a detailed discussion of both optimal control and model-free RL approaches. However, it is quite long and dense, which may make it difficult for readers to identify the key contributions. The discussion of prior work could be more concise, focusing on the gaps that the current method specifically addresses.

2. Overall Logic and Structure of the Introduction
   The introduction mixes several themes, including human motor inspiration, RL challenges, benchmark design, and technical method details. The narrative would benefit from a clearer structure, such as:
   “problem background → limitations of prior methods → core methodological idea → benchmark contribution → main experimental findings.” This would help readers better understand the contributions and novelty.

3. Conceptual Positioning of SkillBlender
   SkillBlender is described as a pretrain-then-blend hierarchical control framework. While inspired by human motor skill development, the introduction conflates hierarchical control, compositional policy methods, and RL algorithms. It is unclear whether SkillBlender is intended as a new RL algorithm, a structured policy composition framework, or a hybrid. This ambiguity undermines the conceptual validity of subsequent comparisons with RL baselines such as PPO or DreamerV3.

4. Reward Design and Generalization Claims
   The introduction emphasizes minimal reward engineering as a key advantage. While the hierarchical structure may reduce reward shaping, the claim that “only one or two reward terms per task” suffice lacks empirical support in the introduction and may be perceived as overgeneralized.

5. Methodological Novelty
   The proposed vectorized weighting mechanism essentially implements a per-joint Softmax mixture, similar in spirit to prior compositional approaches such as MCP. The reported efficiency and stability improvements appear to result primarily from leveraging pretrained skills and simplified rewards, rather than from a fundamentally new reinforcement learning algorithm or principle.

6. Benchmark Contribution
   The introduction of SkillBench is a strength, but its novelty relative to existing benchmarks (e.g., HumanoidBench, ManiSkill3, RoboSuite) may be overstated. Features such as parallel simulation, multi-embodiment support, and motion feasibility are valuable but exist in other platforms. The authors should clearly highlight what is unique in SkillBench, such as specific task designs, evaluation metrics, or assessment protocols that are not addressed elsewhere.

7. Experimental Comparison
   SkillBlender appears not to be a conventional RL algorithm, as the low-level skills are pretrained and frozen while only a high-level controller is optimized. Comparing this system directly with end-to-end RL methods like PPO or DreamerV3 is conceptually imprecise because these approaches differ fundamentally in learning objectives, data requirements, and optimization dynamics.

---

> ### Author Response · Authors · 2025-12-11
> **Response to Reviewer o6cz [1/2]**
>
> Dear Reviewer o6cz,
>
> Thank you for your recognition of our work and valuable comments! Regarding your requested changes, we address as follows:
>
> # 1. Clarity and Scope
>
> Thank you for the valuable suggestion. We will revise the introduction to make it more concise and focused on the key motivation: why existing methods struggle to achieve versatile humanoid loco-manipulation and what specific gaps our method aims to address. These adjustments will be incorporated into the revised manuscript to improve accessibility and clarity.
>
> # 2. Overall Logic and Structure
>
> Thank you for the helpful feedback. Following your recommended structure, we will reorganize the introduction along the sequence: problem background, limitations of prior methods, the core methodology of SkillBlender, benchmark contributions, and main experimental findings. We will also integrate your suggestions and those from Question 1 to refine the narrative and ensure the introduction is presented clearly.
>
> # 3. Conceptual Positioning of SkillBlender
>
> We view SkillBlender as **a novel framework of structured policy combination**. The high-level controller composes goal-conditioned, task-agnostic, and reusable low-level skills in a principled manner. This distinguishes SkillBlender from monolithic RL approaches such as PPO or DreamerV3, and also from hierarchical methods that rely on a single low-level policy or simpler blending mechanisms.
>
> To evaluate the benefit of structured blending, we compare against both monolithic RL baselines and hierarchical approaches such as HB, Sequential, and MCP. These comparisons demonstrate that our hierarchical skill blending mechanism provides improvements over both categories, validating its conceptual and practical advantages.
>
> # 4. Reward Design and Generalization Claims
>
> As documented in Section D of the appendix, all eight high-level tasks in SkillBench are trained using **only one or two** reward terms. This simple design consistently yields successful learning across tasks, supporting our claim that hierarchical skill blending largely reduces the need for extensive high-level reward shaping. We will further clarify this point in the revised introduction.
>
> # 5. Methodology Novelty
>
> First, we want to clarify the comparison with MCP. The original MCP paper uses a small collection of motion clips as its low-level action space and does not employ goal-conditioned, reusable skills, which is itself one of our methodological contributions. For fair comparison, we perform our evaluation under identical conditions by **using the same low-level skills and the same high-level reward terms**. This controlled setting ensures that the performance gains we observe arise from the intrinsic advantages of our **vectorized skill blending mechanism** coupled with the **joint-level Softmax layer**, rather than skills and rewards.
>
> Therefore, our methodological novelty stems from three key components:
>
> a. The integration of **goal-conditioned, task-agnostic and reusable low-level skills**, which provide a structured low-level action space that enables more effective high-level learning;
>
> b. The **vectorized skill blending mechanism** that allows **simultaneous activation** of different body parts for more accurate and flexible policy composition, in contrast to approaches such as Sequential and MCP, which do not support such coordinated composition;
>
> c. The coupled **joint-level Softmax layer** that introduces structured non-linearity and enables smooth composition of physically feasible actions.
>
> The benefit of the low-level skills is demonstrated through comparisons with monolithic methods and HB, while the advantages of the blending mechanism are validated through comparisons with Sequential and MCP. Together, these results support both the novelty and the effectiveness of our contributions.
>
> # 6. Benchmark Contribution
>
> Although parallel simulation and cross-embodiment support exist in some non-humanoid benchmarks, integrating these capabilities into humanoid loco-manipulation benchmarks requires **significant engineering efforts** and is essential for scalable training. Moreover, SkillBench provides eight new humanoid tasks, along with feasibility metrics designed to evaluate motion realism and assess reward hacking, capabilities that existing humanoid benchmarks do not offer [1]. These features collectively support **reproducible and scientifically grounded evaluation** for humanoid whole-body loco-manipulation.

---

> ### Author Response · Authors · 2025-12-11
> **Response to Reviewer o6cz [2/2]**
>
> # 7. Experimental Comparison
>
> For a comprehensive assessment, we evaluate both monolithic RL algorithms (PPO, DreamerV3) and hierarchical baselines (HB, Sequential, MCP). Although monolithic RL differs conceptually from hierarchical RL frameworks, these comparisons serve as **important sanity checks** to illustrate the limitations of simple and flat end-to-end approaches on complex high-level humanoid tasks. Comparisons with hierarchical methods further demonstrate that SkillBlender offers meaningful improvements even within this category, demonstrating the effectiveness of our skill blending mechanism.
>
> - - -
>
> We hope these responses properly address your concerns. Thank you again for your valuable suggestions! We are committed to providing additional responses should you have any further comments.
>
> [1] Sferrazza, Carmelo, et al. "Humanoidbench: Simulated humanoid benchmark for whole-body locomotion and manipulation." arXiv preprint arXiv:2403.10506 (2024).

---

### Review · Reviewer_5nL8 · 2025-11-27

**Summary Of Contributions:**

This manuscript targets humanoid whole-body control and loco-manipulation leveraging, proposing the SkillBlender framework for hierarchical reinforcement learning. This framework performs pretraining on task-agnostic primitive skills, which are dynamically blended to perform high-level tasks.

**Audience:**

Yes

**Audience Explanation:**

Researchers in humanoid fields would find certain values in this study. I hope to see the code and benchmark as open source in the future.

**Claims And Evidence:**

No

**Claims Explanation:**

I think the current version is not sufficient. Please see the Requested Changes below.

**Requested Changes:**

- The main claim is to use a few reward terms in high-level tasks but use various rewards and hand-crafted regularization tuning in low-level tasks. I think that it requires much effort for the tuning and wonder whether this large expense was required in other methods, such as PPO or DreamerV3. Rather than reducing overall reward engineering expense, the proposed method looks like pushing the engineering to low-level to simplify the efforts at high-level. Is it possible to compare the whole engineering effort, including both low-level and high-level?
- The reason to choose blending based on softmax against scalar weight such as MCP is explained with an ablation study, but I would like to request a theoretical analysis or a toy example to provide convincing insights and rationale for this choice.
- Compared with general machine learning, the proposed method should have a distinct, specialized component for humanoid whole-body control as well as loco-manipulation leveraging. This part should be clearly stated.
- The claim of avoiding reward hacking looks interesting, but can this be measured with a quantitative index?
- All experiments are performed in simulation environments such as Isaac Gym, not the real simulation. Furthermore, the comparison was performed for only part of the method in related works, omitting several state-of-the-art methods such as Hover and OmniH2O.
- Is it a typo to omit 4 in the second term of Eq. 7? Similarly, is it correct to omit 5 in the second term of Eq. 10? These reward functions in Appendix D require a short explanation.
- Writing should be improved.
    - comprises of → comprises
    - Architechtures → Architectures
    - Humanoid whole-body control and loco-manipulation remains → Humanoid whole-body control and loco-manipulation remain

---

> ### Author Response · Authors · 2025-12-11
> **Response to Reviewer 5nL8 [1/2]**
>
> Dear Reviewer 5nL8,
>
> Thank you for your recognition of our work and valuable comments! Regarding your requested changes, we address as follows:
>
> # 1. Engineering efforts on high-level and low-level parts
>
> Thank you for the thoughtful comment. While low-level skills require reward tuning, these skills are **task-agnostic and reusable**. Once trained, they can be directly reused and applied to new tasks, allowing the high-level controller to operate with only one or two simple reward terms. Monolithic methods, in contrast, require extensive reward tuning for every individual task, which significantly limits their **scalability to the large variety of tasks** encountered in real-world humanoid applications.
>
> Regarding the overall engineering effort, the low-level component indeed requires some effort to design reward functions, similar to monolithic approaches. However, the high-level component requires only minimal tuning, since it relies on the expressive and reusable low-level skill library. This design principle enables our method to perform well across eight diverse high-level tasks with very limited high-level reward engineering.
>
> Looking ahead, we envision a scalable ecosystem where **robot manufacturers or third-party developers incrementally supply standardized, reusable skills**. For example, companies such as Unitree Robotics are progressively providing firmware updates to their humanoid platforms with new capabilities (e.g., fast running, getting up). Our SkillBlender framework is designed to flexibly incorporate such skills, enabling **compositional generalization** to incrementally unlock more and more complex high-level task capabilities without training from scratch.
>
> # 2. Theoretical analysis on the Softmax layer
>
> Thank you for this excellent question. As we noted in the ablation studies, Softmax introduces **non-linearity** and **enforces structural constraints** on the skill weights by ensuring they lie in the range $[0, 1]$ and sum to $1$. This results in a **convex combination** of skill outputs per joint, where the final action can be intuitively interpreted as a **weighted contribution of each skill**. This non-linearity avoids direct linear combination from neural network outputs to actions, and the constraints discourage the policy from producing arbitrary or extreme outputs, which helps **mitigate reward hacking**, which will happen when removing the Softmax layer.
>
> We also conducted additional ablations in Section G.4 of the appendix, where the joint-level Softmax layer was replaced with joint-level L2 normalization, which is also non-linear. The results show that the L2-normalized variant struggled to learn meaningful behavior, completely failing to blend skills effectively. We attribute this to the fact that L2 normalization only controls the magnitude of the vector but not its structure — it lacks the inductive bias that Softmax imposes, leading to a looser, less structural action space.
>
> # 3. Specialize component for humanoid whole-body control and loco-manipulation
>
> Our work focuses on humanoid loco-manipulation tasks that require coordinated humanoid whole-body behavior. The low-level control structure follows the standard PD controller used in all prior RL-based humanoid systems, where the PD controller maps joint targets from the neural network to torques. We will clearly state this in the revised version of our paper.
>
> # 4. Quantitative index to measure reward hacking
>
> As discussed in the introduction, reward hacking in humanoid control is nuanced and challenging to quantify. In SkillBench, we propose a set of feasibility metrics that serve as practical proxies for assessing the naturalness, stability, and overall realism of humanoid motions. While these metrics are not a strict quantitative index of reward hacking, evaluating them jointly provides insight into which policies are less prone to reward hacking. This multi-metric evaluation provides a more holistic assessment than solely examining task returns, as in prior works [1].

---

> ### Author Response · Authors · 2025-12-11
> **Response to Reviewer 5nL8 [2/2]**
>
> # 5. Comparison with other methods
>
> Thank you for the suggestion. Since all evaluated methods operate in a state-based setting, our comparisons are conducted within simulated benchmarks. In addition to the baselines included in the main paper, we compare our method with HOVER [2] (sparsity mask, joint angle tracking mode) on the H1 BoxPush task. We use the aligned state $s_t$ same as our settings, and train a high-level controller to output the tracker’s input goal for the high-level task. The results are shown below:
>
> |                            | **Error↓**       | Tilt↓ | $h$↑   | $\tau$↓  | $P$↓  |
> |----------------------------|------------------|-----------|----------|---------|---------|
> | **Ours**                   | **0.009±0.007**  | **0.064** | **0.884**   |**15.0**| **9.9** |
> | *HOVER*  | 0.049±0.052      | 0.132     | 0.863 | 20.4    | 12.4    |
>
>
> As shown in the results, our method performs better, consistent with the results against other low-level trackers, which are less performant and still exhibit reward hacking. These results further demonstrate the effectiveness of our skill blending hierarchy.
>
> # 6. Equation explanations
>
> Thank you for pointing this out. The coefficients in the reward terms are not typos. They are tuned values that shape exploration and stabilize learning. Reward shaping is essential in RL, and these coefficients are tuned to encourage effective task learning. This further highlights the advantage of our approach, since the high-level controller requires far less reward tuning compared with monolithic methods, which often require more than ten reward terms for each task.
>
> We also appreciate the opportunity to clarify the meanings of symbols in the equations. The table below summarizes the correspondences:
>
> |   Symbols in equations                         |  Their meanings|
> |----------------------------|------------------|
> | $p$                   | 3D position in space  |
> | $wr$  | wrist      |
> |$bn$|button|
> |$ra$|right arm joint angles|
> |$arti$|cabinet articulation angle|
> |$oriball$|ball in its original state|
> |$xy$|x- and y- portion|
> |$pkg$|package|
> |$\hat{\cdot}$|goal specifier|
>
> We will include these explanations in the revised version of the paper. Thank you again for your suggestions.
>
> # 7. Writing improvement
>
> Thank you for the helpful suggestions. We will correct the identified issues and improve the writing quality accordingly in the revised manuscript.
> - - -
>
> We hope these responses properly address your concerns. Thank you again for your valuable suggestions! We are committed to providing additional responses should you have any further comments.
>
> [1] Sferrazza, Carmelo, et al. "Humanoidbench: Simulated humanoid benchmark for whole-body locomotion and manipulation." arXiv preprint arXiv:2403.10506 (2024).
>
> [2] He, Tairan, et al. "Hover: Versatile neural whole-body controller for humanoid robots." 2025 IEEE International Conference on Robotics and Automation (ICRA). IEEE, 2025.

---

### Review · Reviewer_MoFy · 2025-11-28

**Summary Of Contributions:**

This paper introduces SkillBlender, which is a hierarchical reinforcement learning framework designed for humanoid loco-manipulation tasks. The framework consists of a high-level controller and low-level skills. The high-level controller outputs a subgoal and weight vectors for low-level skills given the current state and task goal. The low-level skills are goal-conditioned policies trained to give actions based on the current state, subgoal. Besides, a new benchmark named SkillBench is also proposed to evaluate the SkillBlender and other baselines. Empirical results demonstrate the effectiveness of the proposed method. The strengths and weaknesses of the paper are as follows:

**Strengths:**
- This paper leverages a hierarchical reinforcement learning framework to address the humanoid loco-manipulation tasks. It decomposes the complex task into high-level controller and low-level skills and pre-trains the low-level skills to improve the learning efficiency.
- The paper is well-written and clearly presents the methodology and experiments.
- The proposed SkillBench benchmark provides a comprehensive evaluation platform for humanoid loco-manipulation tasks.
- Empirical results demonstrate the effectiveness of the proposed method.

**Weaknesses:**
- The motivation for using the hierarchical reinforcement learning framework is not well justified. The advantages of using the hierarchical framework are not clearly explained in the methodology section.
- It lacks a justification for the design of the low-level skills. How to choose the current four skills is not well explained.
- Ablation studies are missing some variants, such as w/o Squatting and w/o Stepping.

**Audience:**

Yes

**Audience Explanation:**

- This paper proposes a hierarchical reinforcement learning framework for humanoid loco-manipulation tasks, which is an interesting and challenging problem in the field of robotics and reinforcement learning.

**Claims And Evidence:**

Yes

**Claims Explanation:**

- Extensive experiments on the proposed SkillBench benchmark demonstrate the effectiveness of the proposed method.

**Requested Changes:**

- Can you elaborate more on the motivation for using the hierarchical reinforcement learning framework?
- Can you provide more justification for the design of the low-level skills? Why did you choose the current four skills? What other low-level skills can be considered?
- Can you provide more ablation studies such as w/o Squatting and w/o Stepping?
- Can you explain why the methods achieve better performance on the hard tasks in Table 3 compared with the medium tasks in Table 2?
- It is better to provide the figures of the HumanoidBench figure in FarReach in Figure 4.

---

> ### Author Response · Authors · 2025-12-11
> **Response to Reviewer MoFy**
>
> Dear Reviewer MoFy,
>
> Thank you for your recognition of our work and valuable comments! Regarding your requested changes, we address as follows:
>
> # 1. Motivation for using hierarchical reinforcement learning framework for humanoid loco-manipulation
>
> Thank you for the question. The motivation for adopting a hierarchical framework arises from two main considerations:
>
> a. From the perspective of human motor development, people learn a repertoire of **task-agnostic, reusable skills** that can be flexibly combined when facing novel tasks. Many tasks share common low-level motor components. By first learning meaningful low-level skills and later composing them, we substantially reduce the need for complex task-specific tuning and enable more robust learning on diverse tasks, since the agent benefits from strong low-level priors. In contrast, monolithic policies often struggle to acquire such versatile capabilities as our experiment results show.
>
> b. Prior RL-based humanoid control methods typically rely on extensive reward shaping to prevent reward hacking and maintain natural behaviors [1]. In our HRL framework, the low-level skills inherently **constrain and regularize** the agent’s actions, which allows the high-level controller to use only one or two reward terms while still avoiding reward hacking and producing naturalistic motions.
>
> # 2. Justifications for low-level skills
>
> In our framework, we welcome all skills that are **task-agnostic, reusable, and physically interpretable**. Generally, skills that do not require precise interaction with specific objects or scene layouts—such as walking, reaching, stepping, squatting, crouching, or bending—fall into this category. These are low-level motor primitives that can be repurposed across a wide range of downstream tasks.
>
> Looking ahead, we envision a scalable ecosystem where **robot manufacturers or third-party developers incrementally supply standardized, reusable skills**. For example, companies such as Unitree Robotics are progressively providing firmware updates to their humanoid platforms with new capabilities (e.g., fast running, getting up). Our SkillBlender framework is designed to flexibly incorporate such skills, enabling **compositional generalization** to incrementally unlock more and more complex high-level task capabilities without training from scratch.
>
> # 3. Ablation studies on Squatting and Stepping skills
>
> To further verify the effectiveness of Squatting and Stepping skills, we conduct experiments on H1 PackageCarry and FootballShoot tasks, where Squatting and Stepping are blended in, respectively. Below we show the experiment results:
>
> H1 PackageCarry (removing Squatting causes complete task failure, as the error is way larger than the success threshold):
>
> |                            | **Error↓**       | Tilt↓ | $h$↑   | $\tau$↓  | $P$↓  |
> |----------------------------|------------------|-----------|----------|---------|---------|
> | **Ours**                   | **0.013±0.008**  | 0.043 | 0.787    | 21.3| 28.9 |
> | *w/o Squatting*  | 0.346±0.094      | 0.018     | 0.780 | 19.7    | 11.6    |
>
> H1 FootballShoot (removing Stepping causes complete task failure, as the error is way larger than the success threshold):
>
> |                            | **Error↓**       | Tilt↓ | $h$↑   | $\tau$↓  | $P$↓  |
> |----------------------------|------------------|-----------|----------|---------|---------|
> | **Ours**                   | **1.109±0.285**  | 0.131 | 0.843    | 26.1 | 92.8 |
> | *w/o Stepping*  | 1.678±0.183      | 0.024     | 0.971 | 13.2    | 25.0    |
>
> As shown in the results, w/o Squatting and w/o Stepping both completely fail the task (they sometimes show numerically better feasibility metrics because the policies become largely inactive and fail to perform meaningful behaviors). The results further confirm the necessity of Squatting and Stepping skills in these tasks.
>
> # 4. Explanations on method performance
>
> Thank you for the question. Our method does not perform inherently better on hard tasks than on medium tasks. Instead, the discrepancy arises because competing baselines fail to learn the hard tasks entirely, leading to much lower performance in those settings. In contrast, our hierarchical approach maintains stable learning and successfully solves these harder tasks.
>
> # 5. Provide figures
>
> Thank you for the suggestion. The reaching skill in HumanoidBench is trained with PPO, and for FarReach this configuration is identical to PPO. In the revised version of the paper, we will include the PPO figure for the HumanoidBench baseline in FarReach.
>
> - - -
>
> We hope these responses properly address your concerns. Thank you again for your valuable suggestions! We are committed to providing additional responses should you have any further comments.
>
>
> [1] van Marum, Bart, et al. "Revisiting reward design and evaluation for robust humanoid standing and walking." 2024 IEEE/RSJ International Conference on Intelligent Robots and Systems (IROS). IEEE, 2024.

---

### Decision · Action_Editor_5CKV · 2026-02-26

**Recommendation:** Reject

**Audience:**

Yes

**Audience Explanation:**

The problem addressed in the paper is highly relevant to the SkillBench benchmark could be also used by many researchers in the field.

**Claims And Evidence:**

No

**Claims Explanation:**

Although the paper is well written and structured, two of the three reviewers found the evidence not convincing. Mostly the concerns remained even after the rebuttal. In short, their nature is prevalence of qualitative explanations over quantitative examples and analyses.

There is an additional standing concern of comparing with not fully compatible methods (PPO and DreamerV3 that need to train from scratch) while not fully comparing with methods of similar nature.